# [Proposal-ML] Training an Agent to find Diamond in Minecraft, MineRL Competition 2021

**Kittaphot Saengprachathanarak** *
Department of Industrial Engineering
Tsinghua University
Beijing, China
zno.ksy@gmail.com
2024400624

**Gausse Mael DONGMO KENFACK** *
Department of Computer Science and Technology
Tsinghua University
Beijing, China
dmgs24@mails.tsinghua.edu.cn
2024403346

## 1 Background

Reinforcement Learning (RL) has successfully been used in many fields with sequential decision-making problems such as games like AlphaGo [8] or in Atari [5]. By integrating human demonstrations, the reinforcement learning agent can be trained to mimic human-like behavior.

MineRL is a Python 3 library that provides an OpenAI Gym interface for interacting with Minecraft, along with datasets of human gameplay [2]. The MineRL competition 2021, centered on the ObtainDiamond task, stands out as a challenging and resource-rich machine learning project. In this task, the agent spawns randomly in different environments, requiring an efficient policy to make optimal decisions, maximize rewards, and ultimately obtain a diamond. This project serves as a strong baseline for human behavior imitation, with potential applications beyond gameplay. It could pave the way for imitating human behavior in real-world settings, which are even more sparse and complex than virtual game environments.

## 2 Definition

The environment is discretized into timesteps $t$. At each timestep $t$, the agent receives a state $s_t$ from the environment and can perform an action $a_t$. Following this, the environment transitions to the next timestep $t + 1$, and the agent receives a reward $r_t$ along with a new state $s_{t+1}$. Since the objective is to find the optimal policy $\pi(s)$, the problem can be formulated as a Markov Decision Process (MDP), represented by the tuple $(S, A, R, T, \gamma)$, where:

- $S$: A set of states representing the Minecraft environment and the agent's inventory.

- $A$: A set of actions, including movement, item crafting, and mining.

- $R$: A reward function, giving positive rewards when reaching specific milestones or obtaining a diamond.

- $T$: Transition dynamics of the environment.

- $\gamma$: Discount factor representing the significance of future rewards.

The objective of the agent is to maximize the expected cumulative reward $\mathbb{E}\left[\sum_{t=0}^{H} \gamma^t R(s_t, a_t)\right]$, where $H$ is the time horizon.

---

*These authors contributed equally to this work

38th Conference on Neural Information Processing Systems (NeurIPS 2024).

# 3 Related Works

The challenge of obtaining a diamond in Minecraft has become a well-established problem, explored extensively through the MineRL competition, which took place in 2019, 2020, and 2021. Numerous teams participated each year, experimenting with a variety of approaches to tackle this complex task [4]. The organizers published the results and strategies of the top-performing teams and an analysis of these strategies reveals that many of them relied on similar methods to leverage human demonstrations in their training processes. Among the approaches used, three main methods consistently emerged as dominant strategies:

- **Imitation Learning** [1] : it seeks to maximize rewards by replicating human trajectories. Typically, a network is trained on demonstration data to predict the correct action based on a given observation, framing imitation learning as a classification problem where each action represents a different class. With this technique an agent can be trained without interacting with the environment.

- **Hierarchical Reinforcement Learning** [3]: this approach focuses on breaking down the overall task into a series of sub-tasks, which is particularly useful in the diamond acquisition task where the agent must complete multiple steps in sequence. By structuring the task hierarchically, the agent can learn distinct skills for each sub-task, gradually progressing toward the ultimate goal of obtaining the diamond.

- **Imitation Learning + Reinforcement Learning**: this approach combines the strengths of both IL and RL. It can involve pre-training the agent on human demonstration data and then continue using reinforcement learning. Alternatively, some sub-tasks may be solved purely through IL, while others are addressed with RL. This hybrid method leverages the benefits of human-like behavior from IL and the adaptability of RL to improve overall performance.

One algorithm that have been used by many teams is Soft Q Imitation Learning (SQIL). Reddy S. [7] stated that it only requires few changes to any standard Q-learning implementation which could be the reason why it was implemented by many competitors. Despite small adjustments to the implementation, it can perform better than Behaviour Cloning especially when any new initial state is introduced.

# 4 Proposed Method

To solve this problem, we propose a method combining Curriculum Learning [6] with Hierarchical Reinforcement Learning (HRL). Our approach focuses on making the agent capable of reaching the diamond from any starting state, by progressively guiding it through essential sub-goals required by the competition, such as collecting wood or crafting tools.

The core of our method lies in a "Top-Down" or "reverse" Curriculum Learning strategy. Instead of training the agent from the initial random states forward, we start by training it near the goal (mining a diamond) and then progressively introduce earlier sub-goals as new starting points. This reverse approach allows the agent to focus on a goal-oriented learning, thereby improving sample efficiency and accelerating convergence.

Our architecture consists of a manager and multiple workers. The manager, trained on human demonstration data [2], is responsible for breaking down the main task into sequential sub-tasks and detect when a sub-goal is reached. Each worker, specialized in a particular sub-task, is trained either through behavior cloning on human demonstrations or through reinforcement learning techniques. As each sub-goal is achieved, the manager assigns the next relevant sub-task to the next worker, enabling smooth progression through the task hierarchy until the diamond is obtained.

This combined use of HRL and reverse Curriculum Learning is designed to increase the agent's adaptability and efficiency in complex, long-horizon tasks, making it robust across a wide range of starting states. By leveraging human data to inform sub-goal sequencing and skill development, our approach aims to achieve optimal performance with minimal environmental interactions.

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
