# OpenReview forum: "[Proposal-ML] Training an Agent to find Diamond in Minecraft, MineRL Competition 2021"
_tsinghua.edu.cn/THU/2024/Fall/AML — THU 2024 Fall AML Submission_

### Official Review · ~Iat_Long_Iong1 · 2024-11-07
**Interesting idea with challenging environment**

**Rating:** 8
**Confidence:** 4

**Review:**

The proposed method combines Curriculum Learning with Hierarchical Reinforcement Learning (HRL) to enhance efficiency in the Minecraft diamond acquisition task. By starting training near the goal and gradually adding earlier sub-goals, the agent can focus more effectively on goal-oriented tasks, improving sample efficiency and speeding up learning. The hierarchical approach allows a manager to assign specific sub-tasks to specialized workers, ensuring smooth progression through the task sequence.

Minecraft's diverse challenges, such as encountering various monsters or lava in caves, pose significant obstacles to achieving the main goal of finding diamonds. Therefore, effectively allocating these challenges to different workers while ensuring the success rate of the main task becomes a crucial focus of this research. The proposal would benefit from a clearer explanation of how the manager's role in defining sub-goals and guiding workers will address these challenges and ensure task success.

---

### Official Review · ~Yufei_Zhuang1 · 2024-11-09
**Interesting and worth-a-try ideas**

**Rating:** 8
**Confidence:** 3

**Review:**

The combination of Hierarchical Reinforcement Learning and "Top - Down" Curriculum Learning with human demonstrations is a smart move. It effectively addresses the complexity of the randomly generated environment and the series of tasks needed to reach the diamond. The approach's focus on increasing sample efficiency and adaptability is promising, especially for handling Minecraft's challenging nature. Overall, it seems well - conceived and has great potential to yield excellent results in the competition.
By combining Hierarchical Reinforcement Learning with Curriculum Learning and incorporating human demonstrations, it can effectively improve the efficiency of sample utilization. Compared with other methods that do not adopt such a combination, it can make better use of the existing data for training.
Utilizing human data and constructing a structured hierarchy of goals makes the training more targeted and can more efficiently move towards the completion of the specific task of "obtaining diamonds". Compared with methods lacking such goal hierarchy construction and utilization of human data, it can be more efficient in accomplishing complex tasks.

---

### Official Review · ~Zijun_Liu2 · 2024-11-09
**Review and Feedback**

**Rating:** 9
**Confidence:** 3

**Review:**

## Overview
The project’s proposed solution to the MineRL contest combines Curriculum Learning with Hierarchical Reinforcement Learning (HRL), aiming to optimize the agent's efficiency by using a “Top-Down” Curriculum Learning approach. Also, the proposal has a comprehensive background and related work analysis, providing a solid foundation for the project.

## Suggestions and Feedback
Overall, the proposal presents a promising approach to training an RL agent to tackle the complex task of obtaining a diamond in Minecraft. I suggest to include more in-depth evaluation metrics, such as convergence speed, sample efficiency, or adaptability across varied starting states, which would enable a more thorough assessment.

---

### Official Review · ~Jinsong_Xiao1 · 2024-11-09
**review for proposal 54**

**Rating:** 9
**Confidence:** 4

**Review:**

This proposal tackles the challenge of training an agent to achieve the diamond acquisition task in Minecraft for the MineRL Competition using a combination of Curriculum Learning and Hierarchical Reinforcement Learning (HRL).

Pros: The proposal aligns with the objectives of the MineRL competition and tackles a widely recognized problem in RL. The use of reverse curriculum learning, combined with hierarchical reinforcement learning, provides a structured, sample-efficient approach to the diamond acquisition problem.

---

### Official Review · ~Ziang_Zheng1 · 2024-11-11
****Review for Paper Proposal: "Training an Agent to find Diamond in Minecraft, MineRL Competition 2021"****

**Rating:** 7
**Confidence:** 3

**Review:**

**Review for Paper Proposal: "Training an Agent to find Diamond in Minecraft, MineRL Competition 2021"**

**Summary:**
The paper proposes a method for training a reinforcement learning agent to find a diamond in Minecraft as part of the MineRL competition. The authors plan to use a combination of Curriculum Learning (CL) and Hierarchical Reinforcement Learning (HRL) to enhance the efficiency and success rate of the agent. They propose a novel “Top-Down” Curriculum Learning approach, where the agent is initially trained near the final goal and then gradually introduced to earlier sub-goals. Their method is structured around a manager-worker architecture that divides the task into sub-tasks, each assigned to specialized agents or “workers.”

**Strengths:**
1. **Novel Curriculum Learning Approach**: The idea of starting training from the goal and working backward is innovative. By focusing on end goals first, the approach has the potential to improve sample efficiency and accelerate convergence.
2. **Clear Problem Definition and Background**: The proposal provides a comprehensive background on the MineRL competition and the challenges of the task. The authors demonstrate an understanding of prior approaches, particularly those that combine Imitation Learning (IL) with Reinforcement Learning (RL).
3. **Strong Motivation and Real-World Potential**: The authors make a compelling case for the broader applications of this approach beyond gaming, such as learning from human demonstrations in more complex real-world scenarios.

**Areas for Improvement:**
1. **Lack of Detailed Evaluation Plan**: The proposal would benefit from a more detailed description of the evaluation metrics and methodologies. While maximizing reward is a goal, it would be helpful to specify how they intend to benchmark their agent against existing methods or measure the effectiveness of each component (e.g., HRL vs. CL).
2. **Clarification of Reverse Curriculum Learning Implementation**: While the reverse curriculum approach is novel, the paper could provide a clearer plan on how this technique will be implemented. How will the agent transition from one sub-goal to the next as the task complexity increases? Providing more insight into potential challenges, such as reward shaping, would strengthen this section.
3. **Discussion on Generalization**: It is unclear how well this method would generalize to scenarios with different environmental configurations or other Minecraft tasks. The proposal could benefit from a brief discussion on how adaptable their approach is for other settings or complex tasks beyond Minecraft.

**Minor Points:**
- Typographical or formatting errors, such as “Aset of actions” and “Arewardfunction,” need to be addressed.
- The explanation of SQIL as a preferred baseline could be expanded with more specifics on why it might be particularly relevant in comparison to other methods.

**Conclusion:**
Overall, this proposal is promising and addresses a complex problem in reinforcement learning with a novel approach. While additional clarification on implementation and evaluation is needed, the method’s potential for advancing curriculum learning strategies in long-horizon tasks is substantial. With further development, this work could significantly contribute to the field of imitation learning and hierarchical reinforcement learning in complex environments.

---

### Official Review · ~Ruowen_Zhao1 · 2024-11-11
**Summary and Concern**

**Rating:** 8
**Confidence:** 3

**Review:**

### Summary
This paper presents a method combining Hierarchical Reinforcement Learning (HRL) with reverse Curriculum Learning to address the MineRL competition’s ObtainDiamond task. By training the agent near the goal (diamond mining) and gradually introducing earlier sub-goals, the approach improves sample efficiency and convergence speed. The proposed architecture employs a manager trained on human demonstration data to assign sub-tasks to specialized workers. The authors present the technical approach with clarity.

### Concern
Relying on predefined sub-goals from human demonstrations could make the agent overly specialized to the sequence provided, limiting its ability to adapt to tasks that require alternate sub-goal paths or dynamic adjustments in real-time.

---

### Official Review · ~Cheng_Gao2 · 2024-11-11
**Review for Training an Agent to find Diamond in Minecraft, MineRL Competition 2021**

**Rating:** 9
**Confidence:** 3

**Review:**

Strengths:

- Very clear task definition.
- The proposed method, which combines Curriculum Learning with Hierarchical Reinforcement Learning, is innovative. The approach of decomposing the task in reverse from the goal should indeed be robust across a wide range of starting states.
- The organization for the agents is also described in great detail.

Weakness:

- A brief introduction to the evaluation metrics and a short overview of other baselines would make this proposal more complete.

Overall, the background and methodology in this proposal are described comprehensively. As I am not familiar with Minecraft, I don't think I can provide further suggestions.

---

### Official Review · ~Ziyu_Zhao6 · 2024-11-11
**Review of "Training an Agent to find Diamond in Minecraft, MineRL Competition 2021" Proposal**

**Rating:** 7
**Confidence:** 3

**Review:**

Overview:
This proposal presents a sophisticated reinforcement learning approach aimed at tackling the MineRL ObtainDiamond task in Minecraft, a well-known benchmark for reinforcement learning and imitation learning methods. The proposed method combines Hierarchical Reinforcement Learning (HRL) with reverse Curriculum Learning to create a scalable, goal-driven training strategy.

Strengths:
	1. Strategic Curriculum Learning: The “Top-Down” or reverse Curriculum Learning approach is innovative, as it starts training from a goal-oriented perspective (near the final goal) and progressively adds earlier sub-tasks. This method improves sample efficiency and accelerates learning by reducing the need to train from random starting points.
	2. Hierarchical Structure: The HRL setup, which divides tasks into sequential sub-goals and assigns each to specialized “worker” agents, promotes modular learning. This allows for a clear and organized approach, enabling each worker to specialize in a particular skill or sub-task.

Limitations:
	1. Complexity in Coordination: Coordinating the manager-worker framework with both HRL and Curriculum Learning increases the complexity of the implementation.
	2. Scalability and Generalization: The proposed method might struggle to generalize to other long-horizon, real-world tasks without substantial adaptation.

---

### Official Review · ~Zou_Dongchen1 · 2024-11-12
**Review for this proposal**

**Rating:** 10
**Confidence:** 4

**Review:**

This is a very interesting proposal. The authors want to train an agent to find Diamond in Minecraft, MineRL Competition 2021.The authors propose to use reverse Curriculum Learning and Hierarchical Reinforcement Learning (HRL) combined approach. The former helps the agent to focus more on goal achievement, while the latter involves managers and workers in a hierarchical approach to task decomposition and completion.
This project is rich in real-world data and therefore is operational. The writing of the proposal is clear and of high quality. The weakness is that the Definition section is not very well articulated with the Proposed Method section. In the Definition section, the author describes the task as a Markov problem, but in the Proposed Method section, the author does not mention this. I hope the authors will explain the relationship between Markov and Reverse Curriculum Learning and Hierarchical Reinforcement Learning (HRL).

---

### Official Review · ~Wanlan_Ren1 · 2024-11-12
**Review for "Training an Agent to find Diamond in Minecraft, MineRL Competition 2021"**

**Rating:** 8
**Confidence:** 4

**Review:**

This proposal presents an effective approach to training an agent to obtain diamonds in Minecraft by combining hierarchical reinforcement learning (HRL) and reverse curriculum learning. The method leverages human demonstration data, enhancing the agent’s ability to learn complex tasks through structured sub-goals, which should improve sample efficiency and accelerate learning. However, the proposed approach may benefit from testing on more complex datasets that require multi-step logical reasoning, which would clarify its adaptability in diverse scenarios. Overall, this approach is promising and could contribute significantly to advancing agent performance in complex, multi-step decision-making tasks in reinforcement learning.

---

### Official Review · ~jin_wang30 · 2024-11-12
**Ambitious and interesting idea**

**Rating:** 9
**Confidence:** 4

**Review:**

Overall, the article clearly demonstrates the ideas and methods of an innovative reinforcement learning project. The article has a high level of problem definition, a review of related research, and the proposed hierarchical learning strategy and reverse curriculum learning strategy, which demonstrates the potential value of the project in the MineRL competition task.

Advantages:
The article is well-structured and gradually introduces the background, problem definition, related research, and proposed methods. In the "Background" section, the article briefly outlines the successful application of reinforcement learning (RL) in the field of games and the goals of the MineRL competition, so that readers can quickly understand the background and importance of the project. The article defines the reinforcement learning problem in the Minecraft environment in detail, expresses the task as a Markov decision process (MDP), and explains concepts such as state, action, and reward. This definition helps to clarify the formal description of the problem and provides readers with a clear task framework. The article also proposes a "reverse curriculum learning" strategy, which enables the agent to learn goal-oriented behavior more efficiently by starting training from a state close to the goal and gradually introducing pre-tasks. This innovative training sequence can improve sample efficiency and accelerate convergence, demonstrating a unique idea for optimizing learning efficiency in complex tasks. The article introduces a "manager-worker" hierarchical structure in the method, allowing each worker to focus on completing a specific subtask, and the manager controls task allocation and subtask detection. This structure can better cope with the complexity of long-term tasks and enable agents to achieve goal-oriented learning in uncertain environments.

Disadvantages:
Although the article mentions the method architecture and hierarchical strategy, it lacks a detailed description of the specific algorithm implementation, especially in terms of the collaboration mechanism between managers and workers, subtask division and scheduling strategies. This may make it difficult for readers to understand the actual operational details and implementation difficulty of the method. In addition, although the article proposes an innovative combination of inverse curriculum learning and hierarchical reinforcement learning, it lacks an analysis of the limitations that these methods may bring. For example, the applicability of inverse curriculum learning in complex environments and the ability to handle sparse rewards are challenges that may be encountered in practical applications.

---

### Official Review · ~Han-Xi_Zhu1 · 2024-11-12
**Review for "Training an Agent to find Diamond in Minecraft, MineRL Competition 2021"**

**Rating:** 8
**Confidence:** 4

**Review:**

The proposal outlines a project to train a reinforcement learning agent to find diamonds in Minecraft as part of the MineRL Competition 2021. The approach combines Curriculum Learning with Hierarchical Reinforcement Learning (HRL), using a "Top-Down" strategy starting near the goal and working backward to earlier sub-goals. The agent is designed to maximize rewards and ultimately obtain a diamond, with potential applications in imitating human behavior in complex, real-world settings.

## Strengths
1. The combination of Curriculum Learning and HRL is a creative strategy that could improve learning efficiency and goal-oriented behavior.
2. Authors give clear definition of the specific task and the task is very interesting.
3. Authors give detailed outline of their methodology. The material is well written.


## Weaknesses
1. The proposal could benefit from a more detailed discussion on how the agent's performance will be evaluated beyond the immediate task of diamond acquisition.